# The distribution of insect pests and the associated loss of stored sorghum in the Kena district of Konso Zone, South-Western Ethiopia

**Ararso Gognsha Desta, Berhanu Hiruy Yeshitila**◉*

Department of Biology, College of Natural and Computational Sciences, Arba Minch University, Arba Minch, Ethiopia

* yeshitilaberhanuhiruy@gmail.com, berhanu.hiruy@gmail.com

**Data Availability Statement:** Yes - all data are fully available without restriction. The data that support the findings of this study are available in OSF repository [osf.io/ud2j4 at https://doi.org/10.

## Abstract

Sorghum is a staple crop grown in the poorest and most food-insecure regions of Ethiopia. But storage insect pests are its key constraints that have been causing considerable loss. Accordingly, an assessment of the prevalence of insect pests and the associated loss of sorghum stored under farmers' traditional storages was done in randomly selected major sorghum-growing kebeles of Kena District using a nested design between November 1, 2019 and December 30, 2020. It was conducted through the sampling of one kg of grain from a total of 360 randomly selected farmers' storages. Pests' abundance was determined by dividing the total number of individual species by the total number of samples. A count and weight method was used to estimate percent grain damage and weight loss by pests. *Sitophilus zeamais*, *Sitophilus oryzae*, *Sitotroga cereallella*, *Tribolium castaneum*, *Tribolium confusum*, *Cryptolestes ferrugineus*, *Cryptolestes pusillus*, *Rhyzopertha dominica*, and *Plodia interpunctella* were the pests identified from grain samples. When compared to mid-latitude, where they were found in the range between 2.36± 0.21 and 17.50±0.98 mean number of species, all of these pests had a considerably (p < 0.05) larger distribution in low-land kebeles, where they predominated in the range between 5.44±2.00 and 23.61±1.75 amean number. During the nine-month storage period, the degree of pest infestation, percentage of grain damage, and weight loss were significantly (p < 0.05) higher in the bamboo bins when no management measures were implemented in the mid-land and low-land kebeles, respectively, than in the barn. Consequently, bamboo bin storage was considered ineffective for sufficient sorghum grain protection against pests and the associated loss over a longer storage period. But barn storage and the use of cultural pest management practices performed better in protecting stored sorghum from pests. Therefore, improving the bamboo bin storage method is needed to improve its grain protection efficacy.

17605/OSF.IO/2VZY3 or at Internet Archive link
https://archive.org/details/osf-registrations-2vzy3-v1.

**Funding:** The author(s) received no specific funding for this work.

**Competing interests:** The authors have declared that no competing interests exist.

## Introduction

Carbohydrate from sorghum and other cereals is the most vital energy fuel for human machines running worldwide. Sorghum, wheat, maize, and rice alone provide about half the calories consumed globally, out of 90% of the world's calorific requirement [1, 2].

Africa is rich in a wide range of grains, including sorghum, maize, teff, millets, and wheat. These grains are an important source of dietary proteins, carbohydrates, vitamins, and minerals for the people on the continent [3]. Sorghum ranks second among cereals and fifth among all crops regarding production in Africa [4]. It is the most resilient cereal crop on the continent in terms of resistance to climate change, flooding, and drought [5, 6]. It is one of the most important and versatile crops grown in sub-Saharan Africa (SSA) [7] that has been used as a food for humans, as feed (forage) for animals, and as an industrial raw material [4].

Ethiopia is the second-largest producer of sorghum in Africa, following Sudan. Sorghum is one of the major staple crops grown in the poorest and most food-insecure regions nationally [8–11]. It is grown at a wide range of elevations, from lowlands to highlands, i.e., in nearly all parts of the country. It is the second most imperative cereal next to maize grown nationally, constituting nineteen percent of the total cereal cultivated and covering about twenty percent of the entire coverage under cereals [8, 12]. Furthermore, Ethiopia is frequently considered the center of domestication due to her possession of the greatest genetic diversity for both cultivated and wild forms [8, 13]. Hence, it is serving to sustain food and nutritional security [8, 11, 12]. Nevertheless, in spite of the aforementioned numerous benefits, the production and storage of sorghum are affected by an array of abiotic and biotic constraints. Among biotic factors, insect pests are revealed to be the most important constraints that cause significant loss during storage [14, 15]. These storage insect pests include *S. zeamais* (Motschulsky) (the maize weevil), *S. oryzae* (Linnaeus) (the rice weevil), *R. Dominica* (Fabricius) (the lesser grain borer), *S. cerealella* (Olivier) (the Angoumois grain moth), *Oryzaephilus surinamensis* (Linnaeus) (the saw toothed grain beetle), *T. confusum* (Jacquelin duVal) (the confused flour beetle), and *T. castaneum* (Herbst) (the red flour beetle) [16, 17]. Loss of grains by insect pests is reported to be in the range of 20–30% in developing countries, while it is indicated to be in the range of 5–10% in developed countries [18]. These grain losses result in not only a physical loss of food grains but also a loss of all the resources used to produce them, including time, labor, land, water, fertilizer, and insecticide, among others [15].

Ethiopian farmers' store their grain in a variety of traditional structures, including mud and thatched roofs, bamboo bins, barns (gotera), wooden walls, and polypropylene and jute bags, for different periods starting from the time of harvesting. Most of these conventional storage facilities are made using materials that are easily accessible in the area. It is reported that most of these traditional farmers' storages offer the ideal setting for the proliferation of storage insect pests and fail to safeguard the grains that are kept under them.

In Kena District (the study area), the majority of farmers prefer to employ barns and bamboo bins among the aforementioned traditional storage options. Grain, including sorghum, has traditionally been stored in cylindrical barns, which are classic storage structures. It is an outdoor storage building with a thatched roof made of dry hay or corrugated iron, and it is constructed from split or whole bamboo poles or other tree sticks. To protect the stored sorghum grain from moisture and pests, it is often raised above the ground with the use of stones or a wooden platform. In order to keep the grain from deteriorating, it also has windows or ventilation holes that let air circulate and lower the temperature and relative humidity in the storage habitat. Bamboo bins are usually made from bamboo splits or twigs. It can be plastered with cow manure from the inside, which can be kept either outside or next to the home wall. Although the majority of farmers use bamboo bins and barns for a

variety of reasons, the efficacy of these conventional storage facilities in protecting grain from pests is debatable.

A variety of pest management practices, such as the use of no management measures, synthetic insecticides, and cultural methods, were also used in the study area. Among these management practices, the most commonly used by farmers was the cultural management method (separating the infested stored sorghum from the non-infested, exposing the infested sorghum to sunlight, mixing sorghum with small cereals like teff and millet, and using botanicals), and the use of no management measures.

Accordingly, understanding the efficacy of commonly used grain handling practices against pests (storage methods and management practices) might have some contribution to the future planning of effective and affordable management strategies in the study area, in particular, and in the country in general. Besides, in-depth research to increase the capacity and effectiveness of the commonly used traditional storage structures and management practices could improve the post-harvest management strategy for a given crop under specific storage conditions. This could also aid in the development of new and improved technologies that the community could adopt [11, 15].

As a result, testing the current status of farmers' traditional grain protection practices against insect pests might provide some insight on how to plan pest intervention strategies. Nonetheless, studies on the prevalence (distribution) of storage insect pests and the accompanying loss in farmers' traditional grain protection practices are scarce and inconsistent in Ethiopia, in general, and the study area, in particular. Accordingly, it has been challenging to develop new and improved grain storage technologies as well as to design and implement insect pest intervention strategies in the nation due to the lack of non-fragmented, documented information. Therefore, the present study was initiated with the aim of determining the prevalence (distribution) of insect pests and the associated quantitative or physical loss of stored sorghum under the commonly used farmers' traditional storage and management practices in relatively different agro-climatic peasant associations in Kena district of Konso Zone of South Western Ethiopia.

## Materials and methods

### The study area and period

The survey was conducted in four major sorghum-growing kebeles, or peasant associations (the smallest administrative unit in Ethiopia) of the Kena District, which is an administrative unit, made up of many kebeles and located in the Konso Zone. These kebeles have comparatively different agro-climates. The peasant associations included Kaho and Kamele (lowlands), as well as the Fasha and Debana (mid-land or mid-latitude) groups. The first two kebeles are situated in a subtropical or sub-humid agro-climate (Woynadega) at a height of more than 1500 meters above sea level, while the last two are in a tropical or sub-arid environment (<1500 m). The study was conducted during the sorghum storage period between November 1, 2019 and December 30, 2020. In the research location, sorghum is only harvested once a year and stored for seven to twelve months by the majority of farmers.

### The study design and sampling procedure

A nested design was used to carry out the study. With the support of the government Ministry of Agriculture (MOA) district and kebele staff, nine villages was randomly selected from each of the aforementioned peasant associations through this design. Then, ten farmers' traditional stores from each village were selected at random to sample the grain. Villages and

representative farmer storages were chosen at random; however, the Zone, District, and Kebeles were chosen purposefully based on the abundance of sorghum production [15, 19].

## Determination of insect pests' prevalence (distribution)

One kilogram of sorghum grain was taken from each of a total of 360 representative farmer stores (i.e., 4 peasant associations x 9 villages x ten farmers' traditional stores from each village) selected using the aforementioned design. The samples were taken from the top, center, bottom, and sides of the storage structures using a hand sampling spear and a sampling scoop. Sampling was done over a nine-month' storage period, during which the stored sorghum grains were more likely to be infested by insect pests. Samples were taken after every fourth farmer's store. When sample grains were not available in formerly sampled storage, the next storage holding grains was considered for sampling.

Samples taken from each farmer's storage in each village of the sampling site were collected in a sampling bag, labeled with relevant information, and kept for further identification of insect pests. Besides, after thorough mixing, the samples from farmers' storages in each village of each kebele were sub-sampled further to come up with a 100-gram sub-sample (i.e., the final sub-sample used for prevalence and the associated physical loss determination). Then, the subsample from each village was sieved using sieves of different mesh sizes for separating the adult insects from the sample grains in the entomology laboratory of Arba Minch University. All alive and dead insects from sub-samples of each village were collected, immediately preserved in 100-ml plastic jars, and kept for identification.

For identification purposes, the keys for identifying insect pests and the instructions found in books [20–24] of stored product insect pests and other arthropods were employed. Identification of insect pests was based on their morphological characteristics, including the shape and type of their antennae, their articulation, type, spines, and color of their wings, the shape of their thorax and legs, the shape of their eyes, the tip of their abdomens, and the shape of their proboscises, among others. Then, insects were sorted according to their orders, families, and species and counted for each subsample grain taken from each village of each kebele in each case, noting the number.

For assessing pests' infestations, the main variables have been abundance (distribution), relative abundance, and Constance (frequency of occurrence) of species found in samples [11, 15, 25–27]. Accordingly, the prevalence (distribution) of pests in the study sites was calculated by dividing the total number of individuals of a species by the total number of samples (in this case, the total kilograms of the grain samples collected). The number of insect pests collected from grain samples of the nine villages of each kebele was grouped (clustered or organized) into three sub-kebele levels (insects from samples of three villages were managed into insects of one sub-kebele sample), and their mean was used to determine the spatial distribution.

## Insect pests' infestation level and the associated percent grain damage and weight loss determination

Insect pests' infestation level and the associated percent grain damage and weight loss estimate (i.e., quantitative or physical grain damage and loss due to insect pests only) were determined only for the commonly used farmers' traditional storages that were under commonly employed pest management practices. I.e., the storage types that have been mostly used by farmers in the study area are bamboo bins and barn storages, in which two commonly practiced pest management measures (cultural measures and the use of no measure) were employed. A variety of storage methods for sorghum were used by framers in Kena district. These include barns or gotera, bamboo bins or unkula, polyethylene sacks in the living house,

store room, woven baskets, clay pots, and underground pits. Of these traditional storage structures, barns (gotera) and bamboo bins (unkula) were the most commonly used by the majority of farmers to store sorghum grain in the study location. A variety of pest management practices, such as the use of no-management measures, synthetic insecticides, and cultural methods, were used in the study area. Among these management practices, the most commonly used by farmers was the cultural management method (separating the infested sorghum from the non-infested, exposing the infested sorghum to sunlight, mixing sorghum with small cereals like teff and millet, and using botanicals), and the use of no management measures. Understanding the efficacy of these commonly used grain handling practices (storage methods and management practices) against pests might have some contribution to the future planning of effective and affordable management strategies in the study area, in particular, and nationally in general. Accordingly, samples from a total of 108 stores of each of the commonly used storage types (108 bamboo bins and 108 barns) were considered for this study.

For determination of pests' infestation level, the mean of insect pest species collected from grain samples of the commonly used storage types under similar management practices and agro-climatic conditions (low land or mid-latitude) was used. In the first place, the number of insect pests collected from grain samples of the commonly practiced storage methods and management strategies of the nine villages of each kebeles (mid-latitude and low-land kebeles) was grouped (organized) into three sub-locality levels. In another way, insects from samples of sorghum stored under the same commonly used grain handling practices (farmers' storage methods and management strategies) in three villages each kebeles were managed in to insects of one sub-locality samples, as mentioned before, and their mean was used for computing the insect pest infestation level.

For estimating the percent of insect-damaged grains and weight loss, one hundred seeds were randomly taken from each grain sub-sample collected from the commonly used farmers' storage and management practices of each kebeles and categorized as damaged and undamaged through a hand lens, and their weight was taken. Holes on grains formed by insect pests were used for separating damaged grains from the undamaged. Damaged kernels were grouped based on size before weighting in comparison to undamaged kernels, and relatively large-sized grains were considered [28]. A triple-beam balance was used for weighting the grains. To reduce the problem of hidden infestation, undamaged grains separated from damaged grains superficially through a hand lens were kept for seven days before a second assessment of grain damage to see whether or not there was an emergence of insect pests from the hidden infestation [29]. Then, the percentage of insect-damaged grains was calculated using the formula indicated below, as adopted by previous researchers [15, 29, 30].

$$\text{Percent of insect−damaged grains (\%)} = \frac{\text{Number of insect−damaged grains}}{\text{Total number of grains}} \times 100$$

Percent weight loss was determined using a gravimetric or count and weight method [15, 30–33].

$$\text{\% loss in weight} = \frac{\text{UNd−DNu}}{(\text{Nd} + \text{Nu})} \text{X } 100, \text{ where}$$

U = weight of undamaged grain, D = weight of damaged grain, Nd = number of damaged grain, and Nu = number of undamaged grain.

Then, following the similar trend of the infestation level determination detailed above, the mean percent of insect damaged grain and weight loss of grain samples collected from commonly used storage types under similar management practices and agro-climatic conditions in

nine villages each kebele were pulled in to the means of three sub-locality levels to estimate the percent of insect damaged grain and weight loss.

## Data analysis

All the data collected from the survey was managed using Microsoft Excel version 2013 and analyzed by the Statistical Program for Social Sciences (SPSS) version 16. Univariate analysis of a general linear model was used to determine insect pests' prevalence (distribution), pests' infestation level, and percent grain damage and weight loss of sored sorghum in a nine-month storage period. The independent variables were the storage types, farmers' pest management measures, and agro-climates of the study sites. The dependent variables were pests' prevalence (distribution), infestation level, and percent grain damage and weight loss. Significant differences between means were compared using Turkey's studentized (HSD) test at P < 0.05. The association between the insect pests' distribution (prevalence) and the different agro-climates of the study site was computed with simple linear regression analysis. Multiple liner regression analysis was used to see the effect of agro-climates, farmers' traditional storage types, and management measures on pests' infestation and the associated loss.

## Results

### The prevalence (distribution) of the major insect pests in stored sorghum

The distribution and the taxonomic position of insect pests of stored sorghum in different agro-climatic peasant associations of Kena district in the ninth month's storage period are indicated in Tables 1 and 2. Pictures of the major pests of stored sorghum are demonstrated in S1 Fig. Accordingly, *S. zeamais*, *S. oryzae*, S. *cereallella*, *T. castaneum*, *T. confusum*, *C. ferrugineus*, C. *pusillus*, *R. dominica, and P. interpunctella* were the major pests of stored sorghum identified from the grain samples. All of these species had significantly (p < 0.05) higher distribution in Kamele and Kaho peasant associations (in lowlands), as they prevailed in the range between 5.44±2.00 and 23.61±1.75 mean number of species per 100g grain sample than in Fasha and Debana (in mid-latitude), in which they occurred in the range between 2.36±0.21 and 17.50±0.98 mean number of species per 100g grain sample.

**Table 1. The prevalence (distribution) of the major insect pests of stored sorghum in different kebeles of Kena district in a nine-month storage period.**

| Insect pest species | Abundance (mean of each insect pest per 100g) in different agro–climatic kebeles | | | | |
|---|---|---|---|---|---|
| | Mid-latitude | | Low land | | In all kebeles |
| | Fasha | Debana | Kaho | Kamele | |
| *Sitophilus oryzae* | 13.89± 1.61[dB] | 15.28±0.64[cB] | 21.67±1.20[bA] | 22.22±1.61[aA] | 18.26±1.37[B] |
| *Sitophilus zeamais* | 16.11±1.13[dB] | 17.50±0.98[cB] | 21.94±0.28[bA] | 23.61±1.75[aA] | 19.79±0.94[A] |
| *Tribolium castaneum* | 12.78±0.43[dB] | 13.89±1.47[cB] | 17.78±1.22[bA] | 21.11±0.56[aA] | 16.39±1.25[D] |
| *Tribolium confusum* | 11.11±1.12[cB] | 11.61±1.00[cB] | 17.22±1.07[bA] | 20.78±1.00[aA] | 15.05±1.05[D] |
| *Cryptolestes ferrugineus* | 6.67±0.17[cB] | 5.56±0.19[dB] | 10.00±0.16[bA] | 11.11±0.48[aA] | 8.33±0.70[E] |
| *Cryptolestes pusillus* | 3.72±0.12[cB] | 3.33±0.16[cB] | 9.72±0.05[bA] | 11.11±0.64[aA] | 6.83±1.11[F] |
| *Rhyzopertha dominica* | 2.78±0.1b[bB] | 2.78±0.16[bB] | 5.56±0.42[aA] | 5.44±2.00[aA] | 4.28±0.39[G] |
| *Sitotroga cerealella* | 13.89±0.41[dB] | 15.56±0.42[cB] | 19.44±0.42[bA] | 22.22±0.33[aA] | 17.78±1.11[C] |
| *Plodia interpunctella* | 2.50±0.25[cA] | 2.36±0.21[cA] | 4.08±0.37[bA] | 5.00±0.15[aA] | 3.52±0.34[G] |

Means with different lower and upper case letters within the row of each kebele and agro-climate, respectively, and upper case letters within the last column are significantly different, p < 0.05% using Turkey's studentized test.

**Table 2. Taxonomic position of the major insect pests of stored sorghum in Kena district.**

| Pest species | Common name | Family | Order |
|---|---|---|---|
| *Sitophilus oryzae* | Rice weevil | Curculionidae | Coleoptera |
| *Sitophilus zeamais* | Maize weevil | Curculionidae | Coleoptera |
| *Tribolium castaneum* | Rust red flour beetle | Tenebrionidae | Coleoptera |
| *Tribolium confusum* | Confused flour beetle | Tenebrionidae | Coleoptera |
| *Cryptolestes ferrugineus* | Flat grain beetles | Cucujidae | Coleoptera |
| *Cryptolestes pusillus* | Merchant grain beetles | Cucujidae | Coleoptera |
| *Rhyzopertha dominica* | Lesser grain borer | Bostrichidae | Coleoptera |
| *Sitotroga cerealella* | Angoumois grain moth | Gelechiidiae | Lepidoptera |
| *Plodia interpunctella* | Indian meal moth | Pyralidae | Lepidoptera |

*S. zeamais* is the most prevalent in all Kebeles grain samples, as it accounted for 19.79±0.94 cumulative mean numbers of species and prevailed in a range between 16.11±1.13 and 23.61 ±1.75 mean numbers of species in a 100-gram grain sample. *S. oryzae* and *S. cereallella*, respectively, were the next predominant species, as they occurred in a range between 19.79±0.94 and 17.52±1.11 cumulative mean number of species per 100g of grain sample in the study area (Table 1). Coleoptera (77.78%) was the most predominant taxa of the insect orders recorded from the stored sorghum grain sample, while Curculionidae, Tenebrionidae, and Cucujidae (33.33%) were relatively more dominant families (Table 2). The linear regression analysis showed a significant relationship between kebeles with relatively different agro-climates and insect pests' prevalence or distribution (Table 3). In addition, the $R^2$ value ranged from 0.64 to 0.97, and thus $\geq$ 64% of the pests' distribution can be explained only by the relative climatic variation of the different study sites (Fig 1). Furthermore, the slope coefficient for the study sites (kebeles) with relatively different agro-climatic conditions was 2.388 (Table 4). Consequently, the relative climatic difference of the different study sites was considered to be a significant predictor of variation in pests' abundance (distribution).

## Pests' infestation level in commonly used farmers grain protection practices

Significantly ($p < 0.05$) higher level of infestation of insect pests (*S. zeamais, S. oryzae, S. cereallella, T. castaneum, T. confusum, C. ferrugineus, C. pusillus, R. dominica*, and *P. interpunctella*, respectively) that ranged between 10.00±1.00 and 34.00±0.58 amen number of each species was recorded in bamboo bin storage in low land agro-climatic kebeles, in which no pest management measure was implemented than in barn storage in nine month storage period. Following a similar trend, significantly ($p < 0.05$) high levels of infestation of the aforementioned insect pests, ranging from 3.67±0.33 to 26.00±1.00, were also observed in bmboo

**Table 3. Linear regression ANOVA table of pests' prevalence (distribution) versus study sites.**

| Model | | Sum of Squares | df | Mean Square | F | Significance |
|---|---|---|---|---|---|---|
| 1 | Regression | 770.019 | 1 | 770.019 | 18.853 | .000[a] |
| | Residual | 4329.389 | 106 | 40.843 | | |
| | Total | 5099.407 | 107 | | | |

a. Predictors: (constant), agro-climate

b. Dependent Variable: Pest prevalence (distribution); *df = degree of freedom.*

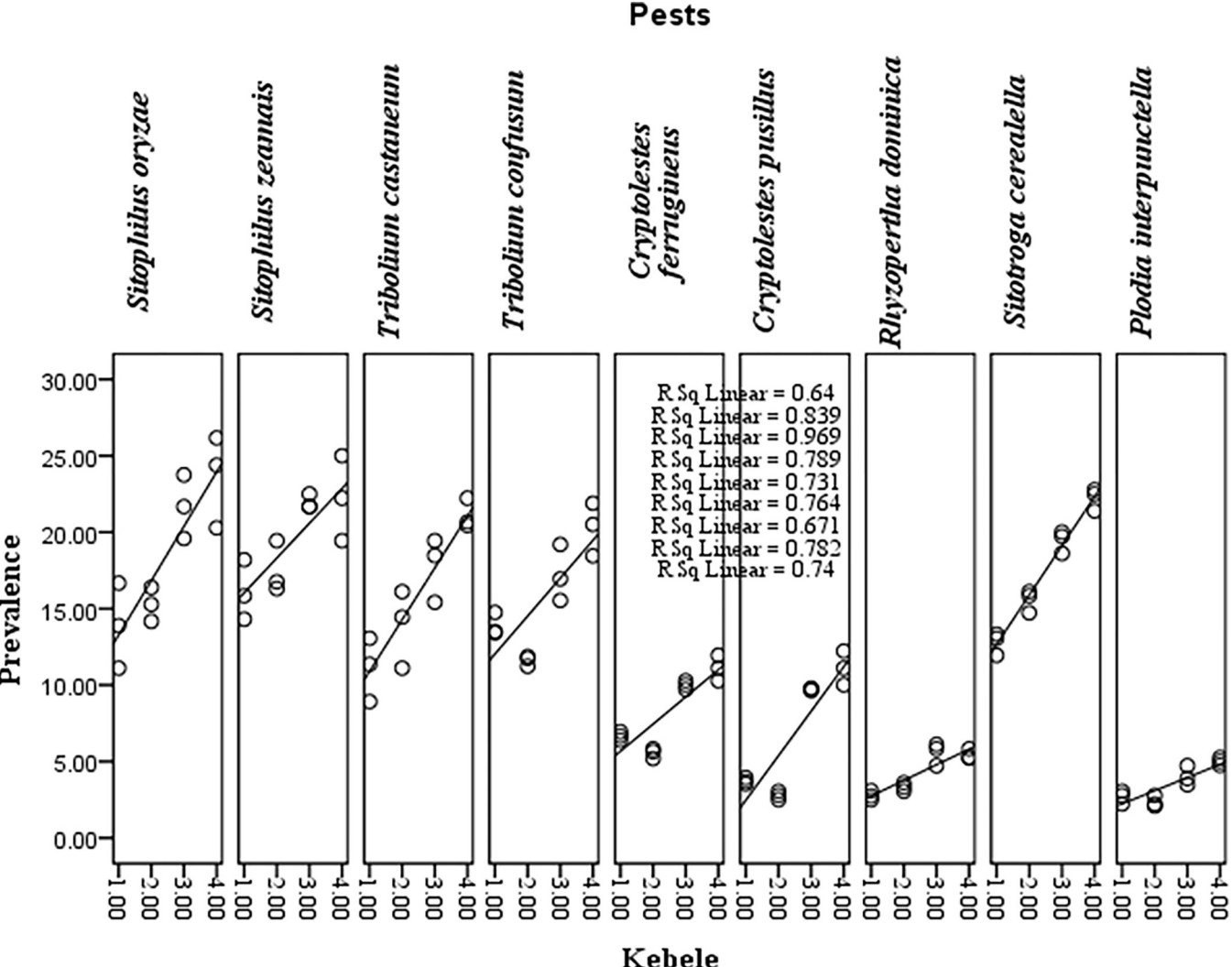

**Fig 1. Linear regression analysis of the association between insect pests' distribution and the different study sites with relatively different climates.**
1 = Fasha, 2 = Debana, 3 = Kaho, and 4 = Kamele. R square = R square; each R square belongs to each pest in its respective order in different sites.

bins than barn storage in mid-latitude Kebeles. *S. zeamais*, followed by *S. oryzae* and *S. cereallella*, were pests that accounted for significantly ($p < 0.05$) the high mean level of infestation that ranged from 16.33±0.88 to 26.00±1.00 and 4.67±0.33 to 8.00±0.58 per 100g grain sample in bamboo and barn storages, respectively, in which no pest management action was taken in mid-latitude kebeles. They had also significantly ($p < 0.05$) higher mean levels of infestation,

**Table 4. Linear regression analysis coefficients of pests' prevalence (distribution) versus study sites.**

| Coefficientsᵃ | | | | | | |
|---|---|---|---|---|---|---|
| Model | | Unstandardized Coefficients | | Standardized Coefficients | t | Significance. |
| | | B | Std. Error | Beta | | |
| 1 | (Constant) | 6.278 | 1.506 | | 4.168 | .000 |
| | Agro-climate | 2.388 | .550 | .389 | 4.342 | .000 |

a. Dependent Variable: Pests prevalence (distribution)

ranging from 24.00±0.58 to 34.00±0.58 and 11.00±0.58 to 15.33±0.88 per 100g grain sample in bamboo and barn storages, respectively, in which no pest management action was employed in low-land kebeles. Therefore, they were considered the most economically important species that accounted for grater infestation and attack of stored sorghum under the commonly used farmers' storages, when no pest management measures were taken (Table 5).

**Table 5. Insect pests' infestation level of stored sorghum under the commonly used grain protection practices of the study area in a nine-month storage period.**

| Agro-climate | kebele | Insect pest species | Infestation level per 100g of the grain sample (Mean± SE) | | | |
|---|---|---|---|---|---|---|
| | | | Bamboo bin | | Barn | |
| | | | No measure | Cultural | No measure | Cultural |
| Low land | Kaho | Sitophilus oryzae | 28.00±0.58[aB**] | 11.00±0.58[cA**] | 13.67±0.33[bB**] | 4.00±0.88[dB**] |
| | | Sitophilus zeamais | 30.67±0.67[aA**] | 11.00±0.58[cA**] | 15.33±0.88[bA**] | 5.67±0.58[dA**] |
| | | Tribolium castaneum | 24.00±1.00[aC**] | 10.33±0.88[cB**] | 12.33±0.33[bC**] | 3.67±1.00[dBC**] |
| | | Tribolium confusum | 23.00±0.58[aD**] | 10.00±1.00[cB**] | 10.00±0.58[bE**] | 3.33±0.587[dC**] |
| | | Cryptolestes ferrugineus | 18.33±0.33[aE**] | 9.00±0.58[cC**] | 9.33±0.33[bF**] | 2.67±0.58[dD**] |
| | | Cryptolestes pusillus | 18.00±0.58[aE**] | 9.00±0.58[cC**] | 9.00±0.58[bF**] | 2.67±0.58[dD**] |
| | | Rhyzopertha dominica | 15.00±0.58[aF**] | 4.00±0.58[cE**] | 8.00±0.58[bH**] | 1.33±0.33[dE**] |
| | | Sitotroga cerealella | 24.67±0.67[aC**] | 10.67±0.33[cB**] | 11.00±0.58[bD**] | 3.00±0.33[C**] |
| | | Plodia interpunctella | 15.33±0.33[aF**] | 6.00±0.58[cD**] | 8.67±0.33[bG**] | 1.33±0.58[dE**] |
| | Kamele | Sitophilus oryzae | 27.67±0.67[aB**] | 11.00±0.58[cB**] | 11.67±0.88[bA**] | 3.67±1.00[dB**] |
| | | Sitophilus zeamais | 34.00±0.58[aA**] | 12.00±1.00[bA**] | 12.00±1.00[bA**] | 5.67±1.00[cA**] |
| | | Tribolium castaneum | 22.00±0.58[aD**] | 10.00±1.00[bD**] | 9.67±1.20[bC**] | 3.33±1.00[cC**] |
| | | Tribolium confusum | 20.67±0.6[7aE**] | 10.00±1.00[bD**] | 9.67±0.67[bC**] | 3.33±1.00[cC**] |
| | | Cryptolestes ferrugineus | 16.33±0.33[aF**] | 7.00±1.00[cE**] | 9.00±0.58[bD*] | 2.33±0.33[dD**] |
| | | Cryptolestes pusillus | 16.00±0.58[aF**] | 5.00±1.00[cF**] | 8.67±0.33[bE**] | 2.33±1.00[dD**] |
| | | Rhyzopertha dominica | 10.00±1.00[aH**] | 3.33±0.33[cH**] | 5.00±1.00[bF**] | 1.33±0.33[dE**] |
| | | Sitotroga cerealella | 24.00±0.58[aC**] | 11.00±1.00[bC**] | 11.00±0.58[bB**] | 3.67±0.33[cB**] |
| | | Plodia interpunctella | 14.33±0.33[aG**] | 3.67±0.33[cG**] | 8.33±0.33[bE**] | 1.33±0.58[dE**] |
| Mid-latitude | Fasha | Sitophilus oryzae | 17.67±0.67[*] | 5.33±0.33[cA*] | 7.00± 1.53[bA*] | 2.33±0.33[dA*] |
| | | Sitophilus zeamais | 22.67±0.67[aA*] | 5.67±0.33[cA*] | 7.00± 1.53[bA*] | 2.67±0.33[dA*] |
| | | Tribolium castaneum | 16.00±0.58[*] | 4.67±0.33[cB*] | 5.00± 0.58[bB*] | 1.67±0.33[dAB*] |
| | | Tribolium confusum | 15.00±1.53[aE*] | 4.00±0.58[cC*] | 4.00± 0.58[bBC*] | 1.33±0.33[dAB*] |
| | | Cryptolestes ferrugineus | 14.33±1.67[aF*] | 3.67±0.33b[cCD*] | 3.00± 1.16[bC*] | 1.00±0.58[cB*] |
| | | Cryptolestes pusillus | 10.67±0.33[aG*] | 3.00±1.15[bD*] | 3.00± 1.16[bC*] | 1.00±0.58[cB*] |
| | | Rhyzopertha dominica | 4.00±0.33[aH*] | 1.33±0.88[bF*] | 2.33±0.58[bD*] | 0.67±0.33[cC*] |
| | | Sitotroga cerealella | 17.00±1.00[aC*] | 4.00±0.58[cC*] | 5.00± 0.58[bB*] | 2.00±0.58[dA*] |
| | | Plodia interpunctella | 7.33±0.33[aG*] | 2.00±0.58[bE*] | 2.67±0.67[bCD*] | 1.00±1.00[cB*] |
| | Debana | Sitophilus oryzae | 19.00±1.00[aB*] | 4.67±0.67[cB*] | 8.00±0.58[bA*] | 2.67±0.33[dAB*] |
| | | Sitophilus zeamais | 26.00±1.00[aA*] | 5.67±0.33[cA*] | 8.00±0.58[bA*] | 3.00±0.58[dA*] |
| | | Tribolium castaneum | 16.67±0.67[aD*] | 4.33±0.33[bD*] | 4.67±1.16[bB*] | 2.33±0.67[cAB*] |
| | | Tribolium confusum | 14.00±1.00[aE*] | 3.00±1.16[cE*] | 5.00±1.00[bB*] | 2.00±0.58[dB*] |
| | | Cryptolestes ferrugineus | 7.00±1.16[aF*] | 2.00±0.33[bE*] | 3.67±0.3[bC*] | 1.33±0.33[cBC*] |
| | | Cryptolestes pusillus | 6.67±0.88[aG*] | 3.33±0.33[bE*] | 3.33±0.33[bC*] | 1.00±0.58[cC*] |
| | | Rhyzopertha dominica | 3.67±0.67[aH*] | 2.00±0.58[bF*] | 2.33±0.33[bE*] | 0.67±0.33[cC*] |
| | | Sitotroga cerealella | 16.33±0.88[aC*] | 3.67±0.33[bC*] | 4.00±0.33[bB*] | 1.67±0.33[cBC*] |
| | | Plodia interpunctella | 4.67±0.33[aG*] | 1.67±0.58[bF*] | 3.00±0.33[cD*] | 1.00±0.58[dC*] |

Means with different lowercase letters within the row and uppercase letters within the column are significantly different, p < 0.05% using Turkey's studentized test. Besides, means with different numbers of asterisks (one (*) or two (**)) within the column for each pest per agro-climate are significantly different (p < 0.05% using Turkey's studentized test).

**Table 6. Multiple linear regression ANOVA table of insect pests' infestation level versus the commonly used farmers grain protection practices and agro-climate.**

| Model | | Sum of Squares | df | Mean Square | F | Significance |
|-------|-----------|----------------|-----|-------------|---------|--------------|
| 1 | Regression | 15898.638 | 4 | 3974.660 | 267.743 | .000a |
| | Residual | 6338.841 | 427 | 14.845 | | |
| | Total | 22237.479 | 431 | | | |

a. Predictors: (Constant), management measure, storage type, agro-climate, intstagclmang

b. Dependent Variable: infestation level; *df = degree of freedom; Intstagclmang = interaction of storage type, agro-climate, and management practices*

Barn storage method along with cultural management measures in mid-latitude Kebeles performs better in protecting stored sorghum against pests, as it is attributed to the mean level of insect pest infestation that ranged between 0.67±0.33 and 3.00±0.58 of the aforementioned insect pests than bamboo bin, in which the infestation ranged from 2.00±0.58 to 6.33±0.33. Barn also had good grain protection efficacy along with cultural pest management practices in low-land kebeles, as it accounted for an infestation level ranging between 1.00±0.58 and 5.00 ±1.00 of the aforementioned insect pests as compared to bamboo bin that attributed to an infestation level in a range of 3.33±0.33 and 12.00±1.00 mean number (Table 5). Multiple linear regression indicated a significantly ($p < 0.05$) higher association between pests' infestation level, commonly used farmer grain protection practices, and agro-climate (Table 6). In addition, the slope coefficient was 11.606 for storage type, -5.018 for agro-climate, -12.921 for farmers' pest management practices, and 1.281 for their interaction effect (Table 7). Accordingly, storage types, farmers' pest management practices, agro-climate, and their interaction were considered to be significant predictors of variation in pests' infestation levels. Furthermore, the $R^2$ value was.712, and thus, $\geq$ 71% of the pests' infestation level can be explained by the interaction effect of relative variation in storage type, pest management practices, and agro-climate of the study sites (Table 8).

## Cumulative pests' infestation level and the associated grain damage and weight loss at district level

The insect pest status determinants (infestation level, percent grain damage, and weight loss) were observed to be greatly influenced by farmers' storage types, pest management measures, and the agro-climate in Kena district. Accordingly, significantly ($p < 0.05$) higher insect pest infestation level that ranging between 12.52±0.29 & 21.89±0.25, and the associated per-cent

**Table 7. Multiple linear regression analysis coefficients of insect pests' infestation level versus the commonly used farmers grain protection practices and agro-climate.**

| Coefficients^a | | | | | | |
|------|-----------|------------------------------|-----------|------------------------------|---------|---------------|
| Model | | Unstandardized Coefficients | | Standardized Coefficients | t | Significance. |
| | | B | Std. Error | Beta | | |
| 1 | (Constant) | 50.329 | 1.773 | | 28.383 | .000 |
| | Storage type | -11.606 | .629 | -.809 | -18.463 | .000 |
| | Agro-climate | -5.018 | .347 | -.782 | -14.468 | .000 |
| | Management measure | -12.921 | .629 | -.900 | -20.555 | .000 |
| | Intstagclmang | 1.281 | .135 | .697 | 9.466 | .000 |

a. Dependent Variable: infestation level; *Intstagclmang = interaction of storage type, agro-climate, and management practices*

b. Dependent variable: infestation. *Intstagclmang = interaction of storage type, agro-climate, and management practices*

**Table 8. Multiple liner regression analysis model of insect pests' infestation level versus commonly used farmers grain protection practices and agro-climate.**

| Model Summary | | | | | | | | | |
|---|---|---|---|---|---|---|---|---|---|
| Model | R | R Square | Adjusted R Square | Std. Error of the Estimate | Change Statistics | | | | |
| | | | | | R Square Change | F Change | df1 | df2 | Significance F Change |
| 1 | .846[a] | .715 | .712 | 3.85293 | .715 | 267.743 | 4 | 427 | .000 |

a. Predictors: (Constant), management measure, storage type, agro-climate, intstagclmang *Intstagclmang = interaction of storage type*, *agro-climate*, *and management practices*

grain damage that ranging between 27.34±0.34 & 49.25±0.43 and weight loss which ranging from 15.84±0.45–28.00±0.58 were recorded in bamboo bin storage than in barn storage type, in which no pest management measure was applied in nine months storage period both in mid-latitude and low land kebeles, the highest which were being in low land kebeles. In contrast, significantly ($p < 0.05$) lower insect pests' infestation level ($\leq 2.96\pm0.49$) and the associated grain damage ($\leq 6.39\pm0.84$) and weight loss ($\leq 3.63\pm0.67$) of stored sorghum were observed in barn storage than in bamboo bins, in which cultural pest management measures were practiced in nine-month storage periods both in mid-latitude and low-land kebeles, the lowest being in mid-land kebeles. Thus, barn storage type along with cultural pest management practices perform better in preserving or protecting stored sorghum grain from pests by keeping it with less infestation level, percent grain damage, and weight loss, particularly in mid-latitude (mid-latitude) kebeles (Table 9).

The multiple linear regression analysis revealed the existence of a significant association between the commonly used farmers' grain protection practices, the agro-climate, and the level of pests' infestation, percent grain damage, and weight loss (Table 10). The slope coefficient of the various independent factors (storage types, farmers' pest management practices, and agro-climate) was $\geq$ -2.88 for pest status determinants (infestation level, % grin damage, and weight loss). Besides, the interaction effect of the storage types, farmers' pest management practices, and agro-climate was highly significant (Table 11). Furthermore, $\geq$ 89.7% of the

**Table 9. Cumulative insect pests' infestation level and the associated percent grain damage and weight loss of stored sorghum in commonly used grain protection practices of kena district in a nine-month storage period.**

| Pest status parameters per 100g | Agro- climate | kebele | Commonly used farmers storage types | | | |
|---|---|---|---|---|---|---|
| | | | Bamboo bin | | Barn | |
| | | | No measure | Cultural | No measure | Cultural |
| Infestation level | Low land | Kaho | 21.89±0.25[**a] | 9.00±0.58[**a] | 10.81±0.81[**a] | 2.96±0.49[**a] |
| | | Kamele | 20.56±044[**a] | 8.11±0.68[**a] | 9.44±0.44[**a] | 2.93±0.07[**a] |
| | Mid-land | Fasha | 13.85±0.22[**b] | 3.74±0.26[**b] | 4.33±0.33[**b] | 1.52±0.24[**b] |
| | | Debana | 12.52±0.29[**b] | 3.15±0.46[**b] | 4.67±0.15[**b] | 1.74±0.26[**b] |
| Percent grain damage | Low-land | Kaho | 49.25±0.43[**a] | 19.40±0.85[**a] | 24.33±0.54[**a] | 6.39±0.84[**a] |
| | | Kamele | 46.26±0.49[**a] | 17.48±0.33[**a] | 21.25±0.45[**a] | 6.31±0.67[**a] |
| | Mid-land | Fasha | 31.17±0.56[**b] | 8.06±0.10[**b] | 9.75±0.38[**b] | 3.33±0.79[**b] |
| | | Debana | 27.34±0.34[**b] | 6.78±0.78[**b] | 10.50±0.29[**b] | 3.75±0.31[**b] |
| Percent weight loss | Low-land | Kaho | 28.00±0.58[**a] | 11.03±0.55[**b] | 13.83±0.45[**ab] | 3.63±0.67[a] |
| | | Kamele | 26.33±0.15[**a] | 9.94±0.06[**a] | 12.08±0.58[**a] | 3.59±0.01[**a] |
| | Mid-land | Fasha | 17.81±0.19[**b] | 4.58±0.28[**b] | 5.54±0.60[**b] | 1.89±0.33[**b] |
| | | Debana | 15.84±0.45[**b] | 3.85±0.15[**b] | 5.97±0.30[**b] | 2.13±0.11[**b] |

Means with two asterisks

(**) within the row and with different letters within the column are significantly different ($p < 0.05\%$ using Turkey's studentized test).

**Table 10. Multiple liner regression ANOVA table of insect pests' infestation level, percent grain damage, and weight loss versus the commonly used farmers grain protection practices and agro-climate.**

| Model | | | Sum of Squares | df | Mean Square | F | Significance |
|---|---|---|---|---|---|---|---|
| 1 | Infestation level | Regression | 1766.623 | 4 | 441.656 | 203.538 | .000[a] |
| | | Residual | 93.305 | 43 | 2.170 | | |
| | | Total | 1859.929 | 47 | | | |
| | % grain damage | Regression | 2733.515 | 4 | 683.379 | 103.180 | .000[a] |
| | | Residual | 284.797 | 43 | 6.623 | | |
| | | Total | 3018.313 | 47 | | | |
| | % weight loss | Regression | 2922.818 | 4 | 730.705 | 204.769 | .000[a] |
| | | Residual | 153.443 | 43 | 3.568 | | |
| | | Total | 3076.261 | 47 | | | |

a. Predictors: (Constant), management measure, storage type, agro-climate, intstagclmang

b. Dependent Variable: infestation level, grain damage, weight loss, *Intstagclmang = interaction of storage type, agro-climate, and management practices*

pests' infestation level, percent grain damage, and weight loss of stored sorghum could be explained only by the relative variation in storage type, management practices, and agro-climate of the study area, as their $R^2$ value ranged from 0.897 to 0.945 (Table 12). Accordingly, storage types, farmers' pest management practices, agro-climate, and their interaction were considered to be significant predictors of variation in pests' infestation level and the associated grain damage and weight loss.

## Discussion

In the current study, a higher prevalence (distribution) of all the major pests is observed in grain samples of the low-land peasant associations than in samples of the mid-latitude (mid-

**Table 11. Multiple liner regression analysis coefficients of insect pests' infestation level versus the commonly used farmers grain protection practices and agro-climate.**

| Coefficients[a] | | | | | | | |
|---|---|---|---|---|---|---|---|
| Model | | | Unstandardized Coefficients | | Standardized Coefficients | t | Significance. |
| | | | B | Std. Error | Beta | | |
| 1 | Infestation level | (Constant) | 44.771 | 3.133 | | 14.290 | .000 |
| | | Storage type | 50.330 | 2.034 | | 24.747 | .000 |
| | | Agro-climate | -11.606 | .721 | -.932 | -16.097 | .000 |
| | | Management measure | -5.017 | .398 | -.901 | -12.614 | .000 |
| | | Intstagclmang | -12.923 | .721 | -1.038 | -17.923 | .000 |
| | % grain damage | (Constant) | 52.438 | 3.553 | | 14.758 | .000 |
| | | Storage type | -12.146 | 1.260 | -.766 | -9.642 | .000 |
| | | Agro-climate | -2.888 | .695 | -.407 | -4.155 | .000 |
| | | Management measure | -15.063 | 1.260 | -.950 | -11.957 | .000 |
| | | Intstagclmang | .872 | .271 | .429 | 3.215 | .002 |
| | Weight loss | (Constant) | 64.702 | .925 | -.927 | -16.061 | .000 |
| | | Storage type | -14.850 | .510 | -.902 | -12.668 | .000 |
| | | Agro-climate | -6.462 | .925 | -1.053 | -18.230 | .000 |
| | | Management measure | -16.856 | .199 | .814 | 8.384 | .000 |
| | | Intstagclmang | 1.669 | .925 | -.927 | -16.061 | .000 |

a. Dependent Variable: infestation level, grain damage, weight loss

**Table 12. Multiple linear regression analysis model of insect pests' infestation level versus commonly used farmers grain protection practices and agro-climate.**

| Model | | R | R Square | Adjusted R Square | Std. Error of the Estimate | Change Statistics | | | | |
|---|---|---|---|---|---|---|---|---|---|---|
| | | | | | | R Square Change | F Change | df1 | df2 | Significance F Change |
| 1 | Infestation level | .975ᵃ | .950 | .945 | 1.47306 | .950 | 203.538 | 4 | 43 | .000 |
| | % grain damage | .952ᵃ | .906 | .897 | 2.57356 | .906 | 103.180 | 4 | 43 | .000 |
| | Weight loss | .975ᵃ | .950 | .945 | 1.88903 | .950 | 204.769 | 4 | 43 | .000 |

a. Predictors: (Constant), storage type, management measure, agro-climate, intstagclmang

Intstagclmang = interaction of storage type, agro-climate, and management practices

land) peasant associations. These differences in the prevalence (distribution) of pests might probably be associated with the difference in climatic conditions of the peasant associations, i.e., since Kamele and Kaho might probably have relatively warmer and humid environmental conditions that could favor and heighten pests' proliferation compared to the rest of the afore-mentioned two peasant associations [19]. Correspondingly, the difference in the climate of the different agro-ecologies is suggested to have its own impact on storage pests' proliferation, spreading, and Constance [15, 28]. Tropical climatic conditions, along with the use of poor storage facilities and poor sanitation of grain storage in SSA, are reported to encourage insect pest attacks [34, 35]. Agro-climatic inconsistency is revealed to affect the severity and timing of outbreaks of insect pests that may change the distribution of the species [36]. Being the most prevalent and important of *Sitophilus zeamais* in grain samples at the study site, followed by *Sitophilus oryzae* and *Sitotroga cereallella*, is in agreement with Hiruy [15] and Tadesse [37], who revealed that these pests are widespread and common in grain storage in Ethiopia.

A greater level of insect pest' infestation and the associated grain damages and weight losses were recorded from the bamboo bin storage method than from the barn (gotera) storage method, in which no management measures were applied in low-land agro-climatic kebeles than mid-land. This result suggests the existence of an effect of variation in temperature and relative humidity in different traditional storages and agro-climates in which sorghum grain was stored on the level of insect pests' infestation. One of the suggested contributors to these efficacy discrepancies between the various farmer storage types is variation in the nature or composition of the farmers' storage types, which in turn may be attributed to their building materials used, construction conditions, and storage places. The other suggested implication is variation in altitude, which could lead to variation in weather conditions that might favor or disfavor pests' multiplication. These might be because of the possibility that this will lead to a drop or an increase in temperature and relative humidity under farmers' grain storages. In a consistent manner, high humidity or moisture coupled with a high temperature is revealed to allow insects and diseases to establish quickly in the storage ecosystem [32, 38]. Compatibly, the dissimilarity in the climate of the different agro-ecologies is suggested to have its own effect on storage pests' propagation, dispersal, and infestation level [15, 28]. The severity and timing of outbreaks of insect pests are also shown to be influenced by agro-climatic inconsistency [36].

The barn storage, along with cultural pest management practices, was observed to have better grain protection performance from pests in the nine-month storage period, particularly in mid-land kebeles. Besides, infestation level, percent grain damage, and weight loss were observed to be greatly influenced by farmer storage types, pest management measures, and the agro-climate in the study area. This result implies the existence of the synergetic influence of storage type, management measures, and agro-climate on pests' infestation level and the

associated grain loss. This might be probably because of the difference in construction materials, type (condition), and place of farmers' traditional storages, as well as the spatial variation that could in turn be attributed to a decrease or increase in temperature and relative humidity that affects pests' survival and reproduction. It might also be due to variations in the efficacy of farmers' pest management practices. Similarly, the type of storage structures used, the duration of storage, and the storage management implemented prior to and during storage are reported to have a great influence on storage losses [15, 28, 32, 39]. Of a variety of storage methods of sorghum used by framers in Kena district (barns or gotera, bamboo bins or unkula, polyethylene sacks, in the living house, store room, woven baskets, clay pots, and underground pits), barns, followed by bamboo bins, were the most commonly used storage methods [S2–S4 Figs]. A barn is a cylindrical, above-ground indigenous storage method used for grain storage. It is made of bamboo or other wooden stem riven, mud, and straw plaster, as well as dry grass (hay) roof covers. It is usually built outdoors at some height above the ground, with windows, doors, and partially mud-plastered openings along its sides to allow ventilation of the stored grain [S2 Fig]. High moisture, relative humidity, and temperature in grain storages in the tropics, including sub-Saharan Africa, are indicated to greatly affect the quality of grain in storage [40]. Various studies have also indicated the existence of a direct relationship between temperature and relative humidity and pest multiplication in storage ecosystems, i.e., as temperature increases, the relative humidity and pests' proliferation also increase [36, 40].

The higher and substantial level of insect pest' infestation and the associated grain damages and weight losses were observed in sorghum stored for nine months under the bamboo bin storage method in low-land and mid-land kebeles, respectively, when no management measure was applied. This implies the infectiveness of the bamboo bin farmers' storage in protection of grains from insect pests' and the associated loss in the longer storage period. Therefore, improving this farmer's traditional storage in a manner that reduces temperature and relative humidity could have paramount importance in preserving stored sorghum for a longer period of time. Different studies also revealed that the farmers' traditional storages in Ethiopia, in particular, and Africa, in general, were not effective against pests' attacks on stored grains and suggested the need for their improvement in order to store grain safely for a longer period of time [8, 15, 40–43]. The aforementioned finding also suggests that the quantity, quality, resources, labor, and food security of poor farmers in the study area were affected. Consistently, Mailafiya et al. [17] revealed that grain losses of even 1% in the various agro-ecological zones can deprive people of a constant supply of quality food (adequate nourishment year-round), income generation, and their means of livelihood. FAO [44] indicated that losses of food and/or food grains, even as low as 5%, should not be ignored, as such physical losses are usually accompanied by qualitative losses that affect the whole mass of the stored grains. Such losses from insect pets are also considered important since they reportedly result in a loss of all the resources necessary to produce food grains, including time, labor, land, water, fertilizer, and insecticide, among others [15].

## Conclusions and recommendations

Regarding the prevalence (distribution) of sorghum insect pests, the degree of infestation, and the resulting loss of stored sorghum, this study offers some insight. In grain samples from lowland peasant associations, the prevalence (distribution) of insect pests is higher than in samples from the mid-latitude, where the surrounding temperature is comparatively cool and humid.

For roughly a nine-month storage duration, barn storage type and cultural pest management practices performed better in protecting stored sorghum grain from pests and the resulting physical loss in both mid-latitude and low-land kebeles. Therefore, it was found that the

integration of barn storage and cultural management techniques was crucial in reducing insect loss in stored sorghum. It is therefore advised to use these methods to control stored sorghum grain pests under farmers' traditional storage conditions in the study area and across the nation. Furthermore, the interaction (synergetic) effect of the different types of storage and farmers' management practices, as well as the agro-climate of the grain storage that is attributed to variation in altitude, was noted to be responsible for the variation in pest infestation level and the corresponding loss of stored sorghum. Hence, for safe grain storage over an extended period of time, these aspects must be taken into account in the development of new and safe technologies as well as in the design and implementation of efficient management strategies that farmers may readily adopt.

## Supporting information

**S1 Fig. Partial views of pictures of the major insect pests recorded from stored sorghum.** (TIF)

**S2 Fig.** Partial views of pictures of barn sorghum storage structure (a-e). (TIF)

**S3 Fig. Partial views of pictures of bamboo bin sorghum storage.** (TIF)

**S4 Fig. Partial views of pictures of pot and pit sorghum storages.** (TIF)

**S1 File. File of letter of support from biology department to kena district authorities.** (PDF)

## Acknowledgments

We authors honestly thank all local district and kebele agricultural staffs for their support in selection of the study sites and data collection.

## Author Contributions

**Conceptualization:** Ararso Gognsha Desta, Berhanu Hiruy Yeshitila.

**Data curation:** Ararso Gognsha Desta, Berhanu Hiruy Yeshitila.

**Formal analysis:** Berhanu Hiruy Yeshitila.

**Investigation:** Ararso Gognsha Desta.

**Methodology:** Ararso Gognsha Desta.

**Validation:** Berhanu Hiruy Yeshitila.

**Writing – original draft:** Ararso Gognsha Desta.

**Writing – review & editing:** Berhanu Hiruy Yeshitila.

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
