## [Decision Letter · Decision Letter 0]

8 Aug 2022

PONE-D-22-13213The Distribution of Insect Pests, the associated Loss of Stored Sorghum and Food Insecurity in the Kena District of Konso Zone, South Western Ethiopia.PLOS ONE

Dear Dr. Yeshitila,

Thank you for submitting your manuscript to PLOS ONE. After careful consideration, we feel that it has merit but does not fully meet PLOS ONE’s publication criteria as it currently stands. Therefore, we invite you to submit a revised version of the manuscript that addresses the points raised during the review process.

We look forward to receiving your revised manuscript.

Kind regards,

Jianhong Zhou

Staff Editor

PLOS ONE

Journal Requirements:

2. We note that Figure 1 in your submission contain copyrighted images. All PLOS content is published under the Creative Commons Attribution License (CC BY 4.0), which means that the manuscript, images, and Supporting Information files will be freely available online, and any third party is permitted to access, download, copy, distribute, and use these materials in any way, even commercially, with proper attribution. For more information, see our copyright guidelines: http://journals.plos.org/plosone/s/licenses-and-copyright.

3. We note that you have stated that you will provide repository information for your data at acceptance. Should your manuscript be accepted for publication, we will hold it until you provide the relevant accession numbers or DOIs necessary to access your data. If you wish to make changes to your Data Availability statement, please describe these changes in your cover letter and we will update your Data Availability statement to reflect the information you provide

Reviewers' comments:

Reviewer's Responses to Questions

**Comments to the Author**

1. Is the manuscript technically sound, and do the data support the conclusions?

Reviewer #1: Partly

Reviewer #2: No

Reviewer #3: Partly

2. Has the statistical analysis been performed appropriately and rigorously? 

Reviewer #1: Yes

Reviewer #2: No

Reviewer #3: Yes

3. Have the authors made all data underlying the findings in their manuscript fully available?

Reviewer #1: Yes

Reviewer #2: No

Reviewer #3: Yes

4. Is the manuscript presented in an intelligible fashion and written in standard English?

Reviewer #1: No

Reviewer #2: No

Reviewer #3: Yes

5. Review Comments to the Author

Reviewer #1: Please see my comments attached. 2. manuscript approach different topics. It has sections on the presence of different plagues, grain damage and type of storage. But the topics are not well connected in the text. It would be good if the authors connect these different themes using a causal/effect to explain the relation. The paper doesn’t do an analysis on food insecurity, so I will suggest removing that from the title of the paper.

Reviewer #2: Manuscript Number: PONE-D-22-13213

Title: The Distribution of Insect Pests, the associated Loss of Stored Sorghum and Food

Insecurity in the Kena District of Konso Zone, South Western Ethiopia

Comments to the authors:

This paper addresses an important topic that is at the heart of the public policies targeting poverty reduction and tackling food insecurity. The data the researchers collected definitely have some potential. Yet, I am wondering about the research gap the researchers are filling; the overall contribution of this paper to the food loss literature is not clear. It should be explained in what the mere measurement of the infestation of pests in sorghum in Ethiopia is contributing to the existing literature. I expand on my comments below:

Major concerns:

- The contribution of this paper to the literature is unclear and a fit in the existing literature is missing: As early as in the introduction, the authors describe the estimated amount of losses in different context and that determining the prevalence of pests is important (but without saying what has been done and what is missing), concluding that with this paper the prevalence would be assessed. It is very unclear what has already been done, what gap in the literature the paper would fill, and how it can contribute to developing more effective interventions. The authors should expand on the introduction or add an additional ‘literature review’/ background section before the ‘material and methods’ section.

- Materials & Methods/ Sampling:

o The research was carried out over 2 years, but at the same time it is said that this is the ‘sorghum storage period’. So, I guess Sorghum is stored the year round? Why then mentioning that this is the specific ‘storage period’? on the same page, you mention that samples were collected over a 9-month period (also repeated later in the paper).. What is the link with the 2 years that are mentioned earlier?

o Clarify concepts of ‘agricultural district and kebele staff’. Who are these people? Are these public employees?

o Watch out for repetitions: paragraphs in lines 91 and 98…

o Better clarify the use of some terms or avoid using them : what is ‘each sampling date? what is the ‘essential information’? what do you mean by ‘complete mixing’: it is unclear what happens to come up with a standard of 100g. What is ‘keys of the books related to stored product insects pest’..? what do you mean by ‘was pulled’ to three sub kebele levels?

- Results:

o Table 1: what do you mean by ‘distribution’? ‘mean weight’ of grains affected by insect pests? %? You don’t weigh the pests I guess. This comments also applies for table 5

o You jump back and forth from summary statistics to regression frameworks (ex. Table 6 are again summary statistics after regressions in tab 4). Start with summary statistics, and then move on to regressions.

o What is the added value of running several simple linear regressions? It is obvious that climate influences pests (I assume that this has been proven by the literature) so it is rather more interesting to see how different types of storage influence pests, by controlling for climate/ kebele fixed effects. So rather, run a multiple linear regression model and control for kebeles, climate and storage type.

o Line 239: it is unclear what you are doing here: Write down the formula for the regression framework. Figure 3 and tables S1, S2 are not easy to find: consider placing them in the main document if you describe them in the text. Lines 248: what do you mean by “1=Bamboo bin storage” and 1.8= Barn storage? What do the numbers indicate.. (no dummies…)

o Line 257: ‘were significant in both the commonly used farmer’s traditional storage’… what are you comparing here? Unclear.

o Significance level notations via ‘lower case letters’ in the tables is very uncommon and difficult to follow. Stick to the **?

o Table 7: variable ‘storage type’ � what type of storage are you looking at ? If this is a categorical variable, you have to transform it into dummies.. or clarify what you are comparing if this is a dummy already. Statements in line 277 are difficult to interpret without the full picture on table 7

o To make this paper more interesting (or even find an added value of this paper), it would be good to explore interactions between climate conditions and storage methods: how do different storage methods perform in different agro-climatic contexts/ changing agro-environmental conditions due to climate change? Try to explore heterogeneity by kebele?

- Discussion:

o Line 319 on: you describe what others have done (also check Delgado et al. 2021 in Applied Economic Perspectives and Policy, that a.o. evaluates teff in Ethiopia). But what is your contribution? What do you add to this existing evidence?

o Lines 343/344: can you conclude this? Are the storage methods that you assess and that perform better associated with increased ventilation?

o Line 352: your results do not really allow you to say anything about the ‘planning management strategies’. The same holds true for ‘awareness creation’ in line 389

Other comments:

- Structure of the article: the use of different size/ formats of the titles and subtitles would facilitate the reading (eg. It is now unclear whether ‘the study design and sampling procedure’ is a subtitle of ‘Materials and Methods’ or at the same level)

- Consider revising the English of the manuscript.

Reviewer #3: A number of suggested improvements for the next version of the paper:

1. The subject of the research is extremely localized (Kena district of the Konso Zone) and external validity of the research is therefore an issue. Some more discussion on implications of the findings for the rest of the country and Africa would be useful.

2. There are some clear contradictions in the paper. They indicate that losses are enormous in the introduction (50% - there is little solid evidence for these findings) while they themselves find only 12%. A major finding in my view is therefore that these losses are so much smaller than is typically assumed and that should best be highlighted.

3. The authors could refer to the global debate on the size of these post-harvest losses as discussed in recent special issue of Food Policy. A relevant finding in the case of Ethiopia is also: Bachewe, F., Minten, B., Taffesse, A.S., Pauw, K., Cameron, A., Endaylalu, T.G. (2020). Farmers’ Grain Storage and Losses in Ethiopia: Measures and Associates, Journal of Agricultural & Food Industrial Organization, https://doi.org/10.1515/jafio-2019-0059

4. The text is confusing regarding the period that is studied exactly. The authors always refer to a nine-period storage period and then refer to the period November 1, 2018 - December 30, 2020, which is 26 months. It would be good if this could be clarified.

5. The authors should be careful in their language related to the results of their regressions. This is an association and does not infer causality. That should be made clearer throughout the text. The different storage structures used by farmers is obviously a choice by them and those that were faced with higher losses might therefore opted for a different storage structure.

6. I understand that quite some farmers in Ethiopia use fumigants and other chemical means to reduce the impacts of pests in these settings. I am surprised that there was no discussion of that issue. This is important and might put doubts on the shown relationships (they could e.g. be more used in modern storage structures and therefore obscure these relationships).

7. Losses can be avoided but there are often costs involved to avoid those, such as improved storage, chemicals, and labor. It would be good if the authors would discuss more costs and benefits of doing so. Maybe farmers prefer having 5% losses as the costs in reducing these losses would be too high. Better information – as suggested in their recommendation - might then not be helpful.

8. The text is sloppy in a number of places and it would be good for the authors to carefully read through the text before it is finalized. For example, there is a paragraph that is repeated in the text (line 298). That is obviously unacceptable. But there are a number of other issues as well.

6. PLOS authors have the option to publish the peer review history of their article (what does this mean?). If published, this will include your full peer review and any attached files.

Reviewer #1: No

Reviewer #2: No

Reviewer #3: No

---

## [Author Response · Author response to Decision Letter 0]

3 Feb 2023

Revision made on reviewers’ comments

First I would like to thank all the reviewers for their critical, constructive, and valuable comments that enrich the manuscript to the scientific standard. I would like also to thank the editors, including the editor in chief for their kind, corrections, supports, and assistance. Below, I tried to indicate revisions that I made for reviewers’ and editor comments with different color.

General reflection or revision made on comments rose

o Green color highlighted revisions in the different parts of thee manuscripts are addressed are addressed for Reviewer 1 comments

o All red color highlighted revisions in the text, tables, and figures among others are addressed for Reviewer 2 comments

o Blue color highlighted revisions in the manuscript are addressed for reviewer 3 comments

o Yellow color highlighted revisions in the manuscript are addressed for editor comments; removing figure 1 & re arranging the sequence of the rest of figures accordingly 

Generally, I tried my best to address the comments raised by reviewers and editor in the whole document or manuscript.

Specific reflection or revision made on comments

sn Commenter Comments raised by editor/reviewers Revision addressed Remark

1

 Editor You may seek permission from the original copyright holder of Figure or remove it Removed, the rest of the figures numbers are rearranged If I am confused in understanding the comment, I am ready to accept any comments that will be raised

 Please ensure that your manuscript meets PLOS ONE's style requirements Tried to check it & ensured I will accept any comments that will be raised regarding this, if it is not sufficiently addressed

 If you wish to make changes to your Data Availability

statement, please describe these changes in your cover letter and we will update your Data Availability

statement to reflect the information you provide Not changed If the already existing one is not correct I will provide any time up on the editor request

3 Reviewer #1: The paper doesn‘t do an analysis on food insecurity, so I will suggest removing that from the title of the paper. removed 

 There is a mismatch between the abstract/introduction, the beginning of the discussion, and the Conclusion. I tried to address it as highlighted with green color in its abstract/introduction, the beginning of the discussion, and the Conclusion. I will accept any comments that will be raised regarding this, if it is not sufficiently addressed

 The different topics are not well connected in the text. It would be good if the authors connect these different themes using a causal/effect to explain the relation. I tried to address it as highlighted with green color in its I will accept any comments that will be raised regarding this, if it is not sufficiently addressed

 The number of farmers (sampling) is not consistent across the text I tried to address it as highlighted with green color in its I will accept any comments that will be raised regarding this, if it is not sufficiently addressed

 It is important to add some definitions to the first part of the paper to clarify the context—for example, bamboo bin storage and barn storage, among others. (I will put more details in the specific comments). addressed as highlighted with green color in its 

 When the authors mention “inappropriate storage facilities” it refers to the “a good condition” of this type of storage? I tried to address it as highlighted with green color by replacing it with better term 

 Discussion. The lead paragraph/section of the discussion should provide analysis or summary of key Results. That will make the subsequent paragraph merger into the paper better tried to address it as highlighted with green color I will accept any comments that will be raised regarding this, if it is not sufficiently addressed

 Are the losses in quantity, quality, or both? I tried to address it as highlighted with green color by explain it as quantitative 

 this one harvest storage period? or many harvest periods I tried to address it as highlighted with green color 

 farmers stores refer to farmer storage or an intermediary? This term is not clear. I tried to address it as highlighted with green color I will accept any comments that will be raised regarding this, if it is not sufficiently addressed

 The total sample was 1 kg? or was 1kg from each farmer store? I tried to address it as highlighted with green color I will accept any comments that will be raised regarding this, if it is not sufficiently addressed

 So, the authors took samples from all the farmers each month? or the samples from farmers were taken over a nine-month period? I tried to address it as highlighted with green color I will accept any comments that will be raised regarding this, if it is not sufficiently addressed

 Line 98 -104: this paragraph is the same as the previous one tried to address it as highlighted with green color I will accept any comments that will be raised regarding this, if it is not sufficiently addressed

 Does the 100 grams refer to grain sample or insects’ sample tried to address it highlighted with green color 

 in previous paragraphs, authors talked about 360 representative farmers storage. This sentence talks about 108 bamboo bins and 108 barns. tried to address it as highlighted with green color through clarification I will accept any comments that will be raised regarding this, if it is not sufficiently addressed

 In the calculation of the percentage of grain damage, are the authors only included the grain damage by insects tried to address it as highlighted with green color through clarification I will accept any comments that will be raised regarding this, if it is not sufficiently addressed

 It should be good to add a description of the climatic characteristics of the study sites to understand the differences in the presence of plagues. Addressed as highlighted with green color 

 It would be good to add a description of these different types of storage under introduction Addressed as highlighted with red color I will accept any comments that will be raised regarding this, if it is not sufficiently addressed

3 Reviewer #2: It is very unclear what has

already been done, what gap in the literature the paper would fill, and how it can contribute to developing

more effective interventions. The authors should expand on the introduction or add an additional ‗literature review‘/ background section before the ‗material and methods‘ section. Addressed as highlighted with red color I will accept any comments that will be raised regarding this, if it is not sufficiently addressed

 The research was carried out over 2 years, but at the same time it is said that this is the ‗sorghum storage

period‘. So, I guess Sorghum is stored the year round? Why then mentioning that this is the specific

‗storage period‘? on the same page, you mention that samples were collected over a 9-month period (also Addressed as highlighted with red color I will accept any comments that will be raised regarding this, if it is not sufficiently addressed

 Reviewer #2: Clarify concepts of ‗agricultural district and kebele staff Addressed as highlighted with red color 

 Better clarify the use of some terms or avoid using them : what is ‗each sampling date? what is the

‗essential information‘? what do you mean by ‗complete mixing‘: Addressed as highlighted with red color 

 Table 1: what do you mean by ‗distribution‘? ‗mean weight‘ of grains affected by insect pests? Addressed as highlighted with red color I will accept any comments that will be raised regarding this, if it is not sufficiently addressed

 You jump back and forth from summary statistics to regression frameworks (ex. Table 6 are again

summary statistics after regressions in tab 4). Start with summary statistics, and then move on to

regressions. Addressed as highlighted with red color If I am confused in understanding the comment, I am ready to accept any comments that will be raised

 What is the added value of running several simple linear regressions? It is obvious that climate influences

pests (I assume that this has been proven by the literature) so it is rather more interesting to see how

different types of storage influence pests, by controlling for climate/ kebele fixed effects. So rather, run a multiple linear regression model and control for kebeles, climate and storage type tried to address it as highlighted with red color If I am confused in understanding the comment, I am ready to accept any comments that will be raised

3 

Reviewer #2:

 Figure 3 and tables S1, S2 are not easy to find: consider placing them in the main document if you describe them in the text. Lines 248: what do you mean by ―1=Bamboo bin storage‖ and 1.8= Barn storage? tried to address it as highlighted with red color If I am confused in understanding the comment, I am ready to accept any comments that will be raised

 were significant in both the commonly used farmer‘s traditional storage‘… what are you

comparing here? Unclear.

o Significance level notations via ‗lower case letters‘ in the tables is very uncommon and difficult to follow. Stick to the **? tried to address it as highlighted with red color If I am confused in understanding the comment, I am ready to accept any comments that will be raised

 variable ‗storage type‘ � what type of storage are you looking at ? If this is a categorical variable,

you have to transform it into dummies.. or clarify what you are comparing if this is a dummy already.

Statements in line 277 are difficult to interpret without the full picture on table 7 tried to address it as highlighted with red color If I am confused in understanding the comment, I am ready to accept any comments that will be raised

 To make this paper more interesting (or even find an added value of this paper), it would be good to explore interactions between climate conditions and storage methods: how do different storage methods perform in different agro-climatic contexts/ changing agro-environmental conditions due to climate change?

Try to explore heterogeneity by kebele? tried to address it as highlighted with red color

It is for this comment that the last table (Table 9.) and the associated result and figures added If I am confused in understanding the comment, I am ready to accept any comments that will be raised

 what is your contribution? What do you add to this existing evidence?

o Lines 343/344: can you conclude this? Are the storage methods that you assess and that perform better associated with increased ventilation?

o Line 352: your results do not really allow you to say anything about the ‗planning management strategies‘.

The same holds true for ‗awareness creation‘ in line 389

Other comments:

- Structure of the article: the use of different size/ formats of the titles and subtitles would facilitate the

reading (eg. It is now unclear whether ‗the study design and sampling procedure‘ is a subtitle of ‗Materials

and Methods‘ or at the same level)

- Consider revising the English of the manuscript tried to address it as highlighted with red color

corrected

removed

corrected

corrected If I am confused in understanding the comment, I am ready to accept any comments that will be raised

4 Reviewer #3 Some more discussion on implications of the findings for the rest of the country and Africa would be useful.

They indicate that losses are enormous in the

introduction (50% - there is little solid evidence for these findings) while they themselves find only 12%.

The authors could refer to the global debate on the size of these post-harvest losses

The text is confusing regarding the period that is studied exact

understand that quite some farmers in Ethiopia use fumigants and other chemical means 

tried to address all as highlighted with blue color

If I am confused in understanding the comment, I am ready to accept any comments that will be raised

 Losses can be avoided but there are often costs involved to avoid those, such as improved storage, chemicals, and labor. It would be good if the authors would discuss more costs and benefits tried to address all as highlighted with blue color If I am confused in understanding the comment, I am ready to accept any comments that will be raised

 The text is sloppy in a number of places and it would be good for the authors to carefully read through thetext before it is finalized. tried to address all as highlighted with blue color If I am confused in understanding the comment, I am ready to accept any comments that will be raised

Best regards!

---

## [Decision Letter · Decision Letter 1]

11 May 2023

PONE-D-22-13213R1The Distribution of Insect Pests and the associated Loss of Stored Sorghum in the Kena District of Konso Zone, South Western EthiopiaPLOS ONE

Dear Dr. Yeshitila,

Thank you for submitting your manuscript to PLOS ONE. After careful consideration, we feel that it has merit but does not fully meet PLOS ONE’s publication criteria as it currently stands. Therefore, we invite you to submit a revised version of the manuscript that addresses the points raised during the review process.

The previous reviewers have re-evaluated the revision and they still have some concerns including the presenting of results and the English language. Please have the comments addressed.

We look forward to receiving your revised manuscript.

Kind regards,

Jianhong Zhou

Staff Editor

PLOS ONE

Reviewers' comments:

Reviewer's Responses to Questions

**Comments to the Author**

1. If the authors have adequately addressed your comments raised in a previous round of review and you feel that this manuscript is now acceptable for publication, you may indicate that here to bypass the “Comments to the Author” section, enter your conflict of interest statement in the “Confidential to Editor” section, and submit your "Accept" recommendation.

Reviewer #2: (No Response)

Reviewer #3: All comments have been addressed

2. Is the manuscript technically sound, and do the data support the conclusions?

Reviewer #2: Yes

Reviewer #3: Yes

3. Has the statistical analysis been performed appropriately and rigorously? 

Reviewer #2: No

Reviewer #3: Yes

4. Have the authors made all data underlying the findings in their manuscript fully available?

Reviewer #2: No

Reviewer #3: Yes

5. Is the manuscript presented in an intelligible fashion and written in standard English?

Reviewer #2: Yes

Reviewer #3: Yes

6. Review Comments to the Author

Reviewer #2: Comments to the authors:

I re-confirm that the paper addresses an important topic that is at the heart of the public policies targeting poverty reduction and tackling food insecurity. The data definitely have some potential. Yet, I still have concerns about the analysis that is carried out and the contribution of this paper.

Comments:

- The fit in the literature has been improved. Still, I think the authors could be clearer from the very beginning on what they do and what their added value is (just as an example, the authors describe the type of storage methods in the discussion – from line 433. This is the core of the paper and should be clear from the very beginning)

- Analysis & results:

o I am puzzled by the way some analyses are carried out and thus the interpretation of the results:

kebele seems to be included as a continuous variable (e.g. Table 4) but these are dummies and should be included as such (interpretation is otherwise impossible). Same for ‘storage type’ what is compared against what? This also does not become clear in the description of the results

o Regression and output tables are still reported in an unconventional way, but maybe this is an issue of a different disciplines. Yet, to get a picture of individual effects of climate versus storage, the analyses should be done jointly. Ideally by also adding interaction effects. Also, how do the authors disentangle between agroclimatic effects and simple kebele effects? Did they combine different kebeles with similar agro-climates?

o Why not including duration of storage or storage management practices in the analysis (mentioned in line 427)? This could have really added some value..

- Discussion:

o A lot of repetitions. As far as I understand there is two main messages:

Lower losses in barn storages (versus bamboo)

Lower losses in highlands

This should be stressed and repetitions avoided.

Reviewer #3: Paper is ok now and the authors have fulfilled the required changes. As there are still some grammar mistakes (I was liner instead of linear), it would be good to give it a last read-through.

7. PLOS authors have the option to publish the peer review history of their article (what does this mean?). If published, this will include your full peer review and any attached files.

Reviewer #2: No

Reviewer #3: No

---

## [Author Response · Author response to Decision Letter 1]

20 Oct 2023

Revision made on reviewers’ comments

First I would like to thank all the reviewers for their critical, constructive, and valuable comments that enrich the manuscript to the scientific standard. I would like also to thank the editors, including the editor in chief for their kind corrections, supports, and assistance. Below, I tried to indicate revisions that I made for reviewers’ and editor comments with different color.

General reflection or revision made on comments rose

o all Green, red and yellow color highlighted parts in the document are the revised parts based on reviwer 2 comments and for the gramer and language problems inprovement

Generally, I tried my best to address the comments raised by reviewer 2.

Comments by reviewer 2

Comment 1. I think the authors could be clearer from the very beginning on what they do and what their added value is (just as an

example, the authors describe the type of storage methods in the discussion – from line433. This is the core of the paper and should be clear from the very beginning)

Response-I tried to address it as indicated in the introduction and methodology

Comment 2. I am puzzled by the way some analyses are carried out and thus the interpretation of the results:

kebele seems to be included as a continuous variable (e.g. Table 4) but these are dummies and should be included as such (interpretation is otherwise impossible).

Response- tis part is also tried to improved

Comment 3. Same for ‘storage type’ what is compared against what? This also does not become clear in the description of the results

Response- tis part is also tried to improved

Comment 4. Regression and output tables are still reported in an unconventional way, but maybe this is an issue of different disciplines. Yet, to get a picture of individual effects of climate versus storage, the analyses should be done jointly.

Response- it is done jointly

Comment 4. Also, how do the authors disentangle between agroclimatic effects and simple kebele effects? Did they combine different

kebeles with similar agro-climates?

Response- yes

Comment 5. Why not including duration of storage or storage management practices in the

analysis (mentioned in line 427)? This could have really added some value

Response- Management practices are included

Comment 5. Discussion:

o A lot of repetitions. As far as I understand there is two main messages:

Lower losses in barn storages (versus bamboo)

Lower losses in highlands

This should be stressed and repetitions avoided.

Response- This also tried to addressed

The reviser or editor can avoid those parts they consider irrelevant or repeated in cases if my response is not satisfactory 

 Best regards!

---

## [Decision Letter · Decision Letter 2]

1 Dec 2023

The Distribution of Insect Pests and the associated Loss of Stored Sorghum in the Kena District of Konso Zone, South Western Ethiopia

PONE-D-22-13213R2

Dear Dr. Yeshitila,

We’re pleased to inform you that your manuscript has been judged scientifically suitable for publication and will be formally accepted for publication once it meets all outstanding technical requirements.

Kind regards,

Abhay K. Pandey

Academic Editor

PLOS ONE

Additional Editor Comments (optional):

Reviewers' comments:

Reviewer's Responses to Questions

**Comments to the Author**

1. If the authors have adequately addressed your comments raised in a previous round of review and you feel that this manuscript is now acceptable for publication, you may indicate that here to bypass the “Comments to the Author” section, enter your conflict of interest statement in the “Confidential to Editor” section, and submit your "Accept" recommendation.

Reviewer #3: All comments have been addressed

2. Is the manuscript technically sound, and do the data support the conclusions?

Reviewer #3: Yes

3. Has the statistical analysis been performed appropriately and rigorously? 

Reviewer #3: Yes

4. Have the authors made all data underlying the findings in their manuscript fully available?

Reviewer #3: Yes

5. Is the manuscript presented in an intelligible fashion and written in standard English?

Reviewer #3: Yes

6. Review Comments to the Author

Reviewer #3: It would be good if the authors could correct the following:

1. What is meant with "key of books" in introduction. It is not clear what that means.

2. It would be good to remove the reference to food insecurity. The authors did not do research on that. Storage losses are different from food insecurity.

7. PLOS authors have the option to publish the peer review history of their article (what does this mean?). If published, this will include your full peer review and any attached files.

Reviewer #3: No

---

## [Editor Report · Acceptance letter]

4 Jan 2024

PONE-D-22-13213R2 

PLOS ONE

Dear Dr. Yeshitila, 

I'm pleased to inform you that your manuscript has been deemed suitable for publication in PLOS ONE. Congratulations! Your manuscript is now being handed over to our production team.

Kind regards, 

on behalf of

Dr. Abhay K. Pandey 

Academic Editor

PLOS ONE